# Co-infection of Dengue, Zika and Chikungunya in a group of pregnant women from Tuxtla Gutiérrez, Chiapas: Preliminary data. 2019

**Leticia Eligio-García**[1], **María del Pilar Crisóstomo-Vázquez**[1], **María de Lourdes Caballero-García**[1], **Mariana Soria-Guerrero**[1], **Jorge Fernando Méndez–Galván**[2], **Sury Antonio López-Cancino**[3], **Enedina Jiménez-Cardoso** [1]*

1 Laboratorio de Investigación en Parasitología. Hospital Infantil de México "Federico Gómez". CdMx. México, 2 Unidad de Enfermedades Emergentes. Hospital Infantil de México "Federico Gómez". CdMx. México, 3 Instituto de Estudios Superiores. Escuela de Medicina, Tuxtla Gutiérrez, Chiapas Mexico

* JimenezCE@yahoo.com.mx

**Data Availability Statement:** All relevant data are within the manuscript.

## Abstract

### Introduction

Dengue, Zika and Chikungunya are RNA Arboviruses present in some areas of Mexico, mainly in the endemic state of Chiapas that is characterized by presence of the vector that transmit them and an ecology that favors high transmission. According to the national epidemiological surveillance system, Dengue has intensified since 2018 and outbreaks continue in various states while for Zika and Chikungunya a decrease in cases has been reported in recent years. The main objective of this study was to determine the incidence of Dengue, Zika and Chikungunya infections during pregnancy in the state of Chiapas.

### Principal findings

The presence of previous and current infections and coinfections diagnosed by molecular (RT-PCR) and immunological (ELISA for IgG determination) techniques indicates a wide circulation of viruses in asymptomatic people, specifically in pregnant women showing that silent infections in dry season contributes to the preservation of viruses.

### Conclusions

From 136 studied samples, 27.7% tested positive for DENV, 8% for ZIKV and 24.1% for CHIKV by RTPCR and the values of IgG in sera show that 83.9% were positive for IgG antibodies against DENV, 65% against ZIKV and 59.1% against CHIKV. Results demonstrated presence of ZIKV and CHIKV, not detected by the epidemiological surveillance system, so the importance of establishing proactive epidemiological systems more strict, especially because these infections in pregnant women can cause severe health problems for newborn children.

**Funding:** This research work was financed with federal funds. No. HIM-2017-020-SSA-1311. The funders had no role in study design, data collection and analysis, decision to publish, or preparation of the manuscript.

**Competing interests:** The authors have declared that no competing interests exist.

## Author summary

A study was conducted in a group of pregnant women from a region of southern Mexico where Dengue, Zika and Chikungunya are endemic to determine the presence of three viral infections, clinical manifestations and congenital damage in newborns. Epidemiological surveillance of diseases is mainly based on the incidence of these diseases; however, asymptomatic carriers also play an important role in disease transmission and are generally not included in the systems. In the case of the three arbovirosis studied, asymptomatic infections can occur until 60–70% of cases and play an important role in the disease transmission and level of protection of population. The three arboviruses, transmitted by the bite of the *Aedes aegypti* mosquito, in the same endemic scenario have the possibility of producing a high rate of asymptomatic infections and coinfections, which may go unnoticed. The study provides valuable information to improve knowledge of transmission in a period of low transmission (dry season) and also to understand that silent transmission has medical importance, especially in pregnancy, due the congenital damage that can cause to newborn. Therefore, it is important to design strategies to incorporate information into the national and international epidemiological surveillance system.

## Introduction

Dengue (DENV), Zika (ZIKV) and Chikungunya (CHIKV) viruses are RNA arboviruses that have reemerged from outbreaks in Oceania and in recent years have been dispersed as emerging diseases in the countries of America [1]; DENV and ZIKV belong to the Family Flaviviridae, genus Flavivirus and CHIKV is from Togaviridae family, genus Alphavirus. DENV are widely distributed throughout the world in tropical and neotropical areas and ZIKV and CHIKV have recently been introduced in America and they are spreading in areas where DENV is endemic. Dengue and Zika diseases are very similar in their benign form [2], although Dengue can develop a serious disease and Zika causes in some cases congenital malformations and neurological illness, as well as a large number of asymptomatic infections [3–4].

The main symptom of the disease caused by CHIKV are arthritis and neurological disease and in many cases asymptomatic infections; however, the three viruses (Fig 1) can be confused frequently [5–6].

DENV, ZIKV and CHIKV are transmitted mainly by the bite of the *Aedes aegypti* mosquito, although it is also suspected *that Aedes albopictus* may participate too. Due to vectors that transmit DENV has not been controlled, it was expected that the other viruses follow the same trends and endemic areas [7]

At the international level, DENV, ZIKV and CHIKV have a simultaneous circulation among endemic countries, having great difficulty to be studied because laboratory studies are required to be able to differentiate not only between these three viruses but also between many other febrile pictures [8]

In Mexico, Dengue infection reappears in 1978 by the coast region of Chiapas and in 1979 by Quintana Roo after being absent since 1961 [9]. The first indigenous case of CHIKV is detected in September 2014 and ZIKV in October 2015 [10]. These three arboviruses can be considered endemic in the country and share the same endemic areas. The endemic behavior of DENV varies from epidemic years and others with low transmission; however, it is not possible to establish an epidemiological behavior for CHIKV and ZIKV because the

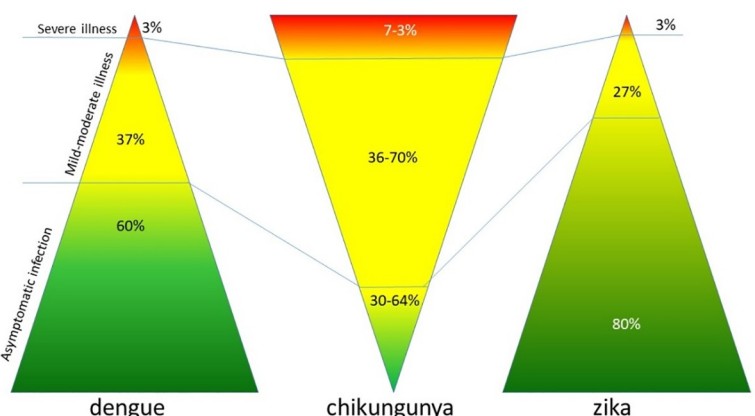

**Fig 1. Differences in clinical manifestations of viral infections.**

epidemiological surveillance system of the Ministry of Health of Mexico established a rule that prevents it fully [11]. Hence the importance of developing studies to know the incidence and prevalence of DENV, ZIKV and CHIKV.

Globally, several studies have been carried out to determine the magnitude of Dengue and Zika, its study is complex since they belong to the same Flavivirus family [12–15]. It has been observed that the prevalence of primary infections by DENV increases with age. Nicaraguan reports show that in a cohort of children, positive serology increases from 22% at 2 years to 95% at 9 years; While in Singapore, where significant progress has been made in vector control, the increase in seropositivity is related to the age of the people, and in the adult population, which has suffered from various infections, the reported data is underestimated [16–17]. In Mexico, studies on the prevalence of infections show that in the state of Yucatan this increases with age and that between two studies carried out in 1996 and 2006, a prevalence of 59% and 82% respectively was found, that is, an increase annual of 2%. Several seroprevalence studies have been carried out in different years and places in Mexico, finding high percentages of antibodies from previous infections, mostly prevalence of immunological memory IgG antibodies greater than 50% [18].

According to the national epidemiological surveillance system, Dengue has intensified since late 2018 and outbreaks continued in several states in 2019. In this period of time, 23,713 cases of dengue fever and 10,786 cases of Dengue with more serious clinical signs were reported. However, 860 cases of Zika were reported in 2018 and 50 in 2019. The reported cases of Chikungunya in 2018 were 39 and in 2019 only 9 [19–21]. This means that those ZIKV and CHIKV are already significantly decreased according to the national system notified by the weekly epidemiological bulletin [22].

In the state of Chiapas for 2018, the national system reported a total of 760 cases of dengue fever and 2,599 of severe dengue. For 2019, reports through week 46 are 581 cases of dengue fever and 972 of severe dengue. According to epidemiological behavior, the season of greatest transmission is between the months of May and November [23].

The association of adverse fetal outcomes of viral infections is not yet clear; DENV has been associated with postpartum hemorrhage. ZIKV is a teratogenic infectious agent associated with severe brain and central nervous system injury, fetal growth restriction, placental insufficiency, and fetal death, while CHIKV may be associated with severe neonatal infections and long-term morbidity [24–26].

This information is epidemiology relevant and the objective of this study was to determine the relationship of DENV, ZIKV and CHIKV infections during pregnancy in the state of

Chiapas from the point of view of the incidence, taking into account that the official epidemiological figures of probable cases do not match the positive cases. (Fig 2).

## Methodology

Ethics Statement: The ethical considerations were following in accordance with the Institutional committee of Children Hospital that reviewed and approved the project (HIM-2017-063). Written consent and assent was obtained from all participants, which was signed either by themselves or a responsible family member.

## Biological samples collection

From February to august 2019, one hundred and thirty six blood samples were obtained from pregnant woman who lived in the central region of the state of Chiapas, Mexico and who attended their prenatal check-up or their delivery to the General Hospital of the Ministry of Health in the city of Tuxtla Gutierrez, in the state of Chiapas.

Their ages ranged from 14 to 43 years old, 12 of them were under 18 years old, the youngest being 14 years old. One of them was 26 weeks of pregnancy, 10 woman was 32–35 weeks, 80 was 36–39 weeks and 35 was in week 40. Clinical data was gathered by asking to the patients about their physical, clinical and epidemiological characteristics.

Five mL of blood were obtained by venous puncture and put into assay tubes containing no anticoagulant. After collection the blood was allowed to clot by leaving it undisturbed at room temperature, during 15–30 minutes. The clot was removed by centrifuging at 1,000–2,000 xg for 10 min and the resulting supernatant, serum, was carefully removed using a Pasteur pipette. Samples were aliquot, labeled and store at –20˚C until use [27].

Serums donated by IMSS obtained from positive patients with confirmed diagnosis of each virus were used as positive control.

## Extraction of Viral RNA

Total RNA is extracted from 140mL serum samples by using the Favor Prep Viral Nucleic Acid extraction kit (No. FAVNK 001–2. Favorgen Biotech Corp, Tw) according to the

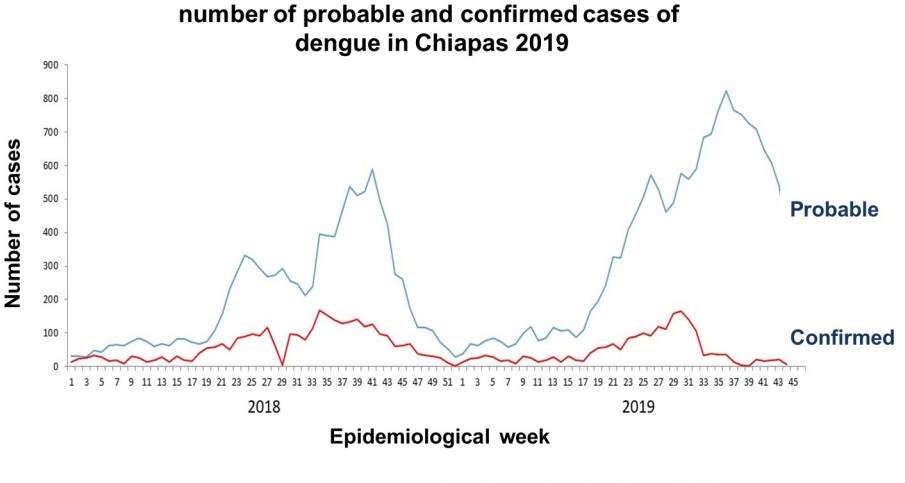

**Fig 2. Number of probable and confirmed cases of Dengue in Chiapas 2019.** Until the epidemiological week 45. From: Unique information platform (SINAVE).

manufacturer's instructions: The RNA is extracted in 60μL of elution buffer and is used immediately to synthesize the cDNA and amplify by One step RT-PCR.

## One step RT-PCR kit for DENV, ZIKV and CHIKV

To identify Dengue, Zika and Chikungunya the RealStar One step RT-PCR Kit 1.0 (DENV No. 282013, ZIKV No. 591013 and CHIKV No. 012013. Altona Diagnostics, Ger) for each one was used [28]. The test consists of three processes in a single tube assay: Reverse transcription of target and Internal Control RNA to cDNA, PCR amplification of target and Internal Control cDNA and simultaneous detection of PCR amplicons by fluorescent dye labeled probes. Master A and Master B reagents contain all the components (PCR buffer, reverse transcriptase, DNA polymerase, magnesium salt, primers and probes) and they were mixed with 9.0μL of extracted RNA in a 15μL reaction volume. Reaction mix was performed in a thermo cycler CFX96 Real Time System (No. 1855095–IVD Bio Rad) using Rox as fluorophore calibrator, FAM like reporter to RNA virus and Joe as reporter of internal control. The amplification program consisted of a 20-minute reverse transcription step at 55°C and then 2 minute for Taq polymerase activation at 95°C, followed by 45 cycles of amplification at 95°C for 15s, 55°C for 45s, and 72°C for 15 s.

A positive sample was plotted with the resulting RFU values (relative fluorescence), to classify the values analogously to viral load. Analysis data was made with the software CFX Manage [29–30].

## Detection of antibodies IgG against DENV, ZIKV and CHIKV by ELISA [31–33]

Anti-DENV, Anti-ZIKV and anti-CHIKV IgG ELISA kits (EUROIMMUN; EI-266a-9601-G, EI-2668-9601G and EI-293a-9601G) were used as recommended by the manufacturer to determine the level of IgG antibodies in serum of pregnant women. Microtiter plates coated with recombinant DENV-NS1, ZIKV-NS1 and CHIKV respectively were used to perform the assay. Briefly, 100 μL of serum diluted 1:100 in sample buffer were added to wells and allowed to react for 60 min at 37°C. Thereafter the wells were washed three times with wash buffer. Bound antibodies were detected by applying rabbit anti-human IgG peroxidase conjugate for 30 min at room temperature, and then wells were washed three times with wash buffer, followed by staining with tetramethylbenzidine for 15 min. The enzymatic reaction was stopped by addition of 100 μL 0.5 M sulphuric acid. Three calibrators as well as positive and negative controls were provided with the test kit and assayed with each test run. Color intensity of the enzymatic reactions was determined photometrically at 450 nm resulting in extinction values. A signal-to-cut-off ratio (extinction sample/extinction calibrator) was calculated for each sample. To ensure high specificity, the borderline range ($\geq 0.8$ to $< 1.1$) was established between the highest negative and the lowest positive validation sample, resulting in a positivity cut-off of $\geq 1.1$.

## Results

The studies for One step RT-PCR were surprising since they show ongoing asymptomatic infections and of the 136 studied samples, 27.7% tested positive for DENV, 8% for Zika and 24.1% for Chikungunya (Table 1). Only two pregnant women had a fever in the two weeks before the sample was taken, four had a fever between 1–2 months before the sample was taken. Only 5 pregnant women had a rash during pregnancy and none of them tested positive for RT-PCR for ZIKV, but two tested positive for DENV and three for CHIKV. The other symptoms were not relevant for the study.

**Table 1. Results of One step RT-PCR of pregnant women and incidence of DENV, ZIKV and CHIKV according to age.**

| Age group | total | DENV | % | ZIKV | % | CHIKV | % |
|---|---|---|---|---|---|---|---|
| <20 years | 33 | 8 | 23.5 | 3 | 8.8 | 10 | 29.4 |
| 20–24 | 41 | 15 | 36.6 | 5 | 12.2 | 9 | 22.0 |
| 25–29 | 34 | 7 | 20.6 | 2 | 5.9 | 5 | 14.7 |
| 30–34 | 16 | 4 | 25.0 | 1 | 6.3 | 4 | 25.0 |
| 35 and + | 12 | 4 | 33.3 | 0 | 0.0 | 5 | 41.7 |
| total | 136 | 38 | 27.7 | 11 | 8.0 | 33 | 24.1 |

By analyzing and plotting the relative fluorescence values (RFU) in each amplification and extrapolating it with a graphic of positive control amplification at different concentrations, it is possible to establish numerical values, that can be used as an analogy of the viral load, in this way a classification was obtained among the positive samples, such as low, medium and high viral concentration (Fig 3).

By this same technique, 11 samples of pregnant women with co-infections positive for DENV and CHIKV, 3 for DENV and ZIKV, 2 for ZIKV and CHIKV and 2 positive for DENV, ZIKV and CHIKV were identified (Fig 4).

In the other hand, the values of IgG in sera for current or recent infection (possibly within the last 12 months) show that 83.9% were positive for IgG antibodies against DENV, 65% against ZIKV and 59.1% against CHIKV (Table 2). These data are within the range of prevalence of antibodies observed in other endemic regions of Mexico for these age groups. We do not have an explanation why a low prevalence of IgG was observed for dengue fever 29% and for Chikungunya 22% in the 20–24 year age group of pregnant women, since an increase in the prevalence is expected at an older age, due to repeated virus infections.

## Discussion

The results obtained in this research show some relevant and important considerations to take into account about the epidemiology of Dengue, Zika and Chikungunya in pregnant women.

During 2019, the information notified by the national epidemiological surveillance system for the state of Chiapas, reported the symptomatic cases of DENV, and no cases of ZIKV and CHIKV. For this reason the presence of current infections diagnosed by molecular techniques (RT-PCR) is very relevant, since it indicates a wide circulation of viruses in asymptomatic people, specifically in pregnant women, which was the study group, suggesting that the infection remains sub diagnosed.

By RT-PCR analyses, the presence of coinfections between the three viruses was observed; the most frequent was DENV and CHIKV with 11 cases of positive pregnant women, indicating the endemicity of the three febrile diseases and the fact that the clinician is not enough for a diagnosis differential. In the case of DENV and ZIKV, both have a crossed immunological reaction, so in cases of coinfections the molecular diagnosis is useful to rule out the cross reaction and avoid false positives [34]. The widespread emergence of DENV and increase in CHIK and DENV cases warrant the need for more effective surveillance to monitor the spread of these deadly arboviruses so that timely control strategies can be implemented [35].

The serum samples from pregnant women included in this study were collected and evaluated between February and August 2019, that is, after the months of high transmission of 2018 and at the beginning of the interepidemic period of low transmission. It is relevant that 46.3% of women are positive for one or more of the three studied viruses by RT-PCR because it represents a high presence of circulating viruses, especially in asymptomatic people [36]. These

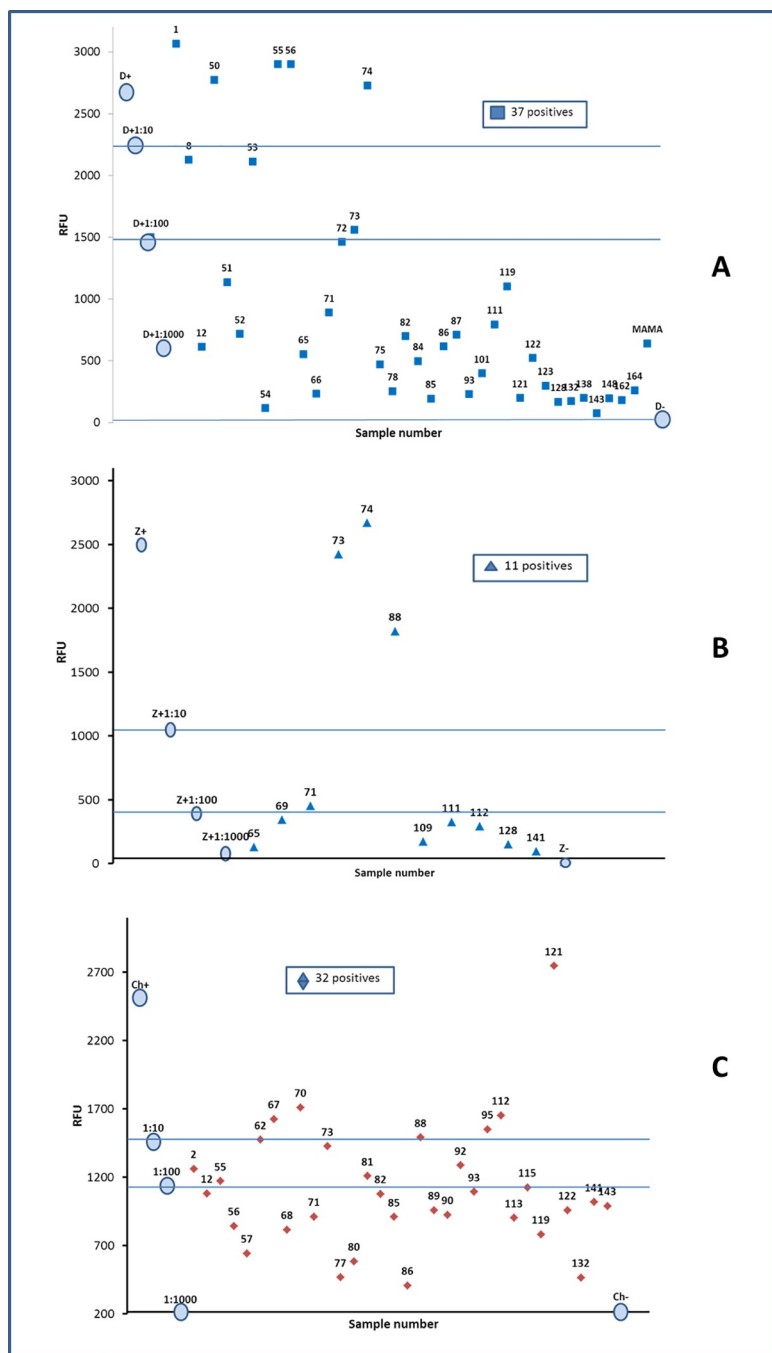

**Fig 3. One step RT-PCR. RFU values to identify Dengue (A), Zika (B) and Chikungunya (C) in sera of pregnancy woman from Chiapas.** The data allow to classify the samples according to the level of fluorescence, and to establish an analogy with the viral load in low, medium or high. The circulated data correspond to the curve of the positive control.

data show that the presence of viruses in relatively low transmission times is relevant and contributes to the preservation of viruses in periods of low interepidemic transmission.

Although the pregnant women included in the study represent a very small proportion of the population of the city of Tuxtla Gutierrez and its surroundings, they were included at random, that is, those who attended the service on the day that coincided in the search for subjects

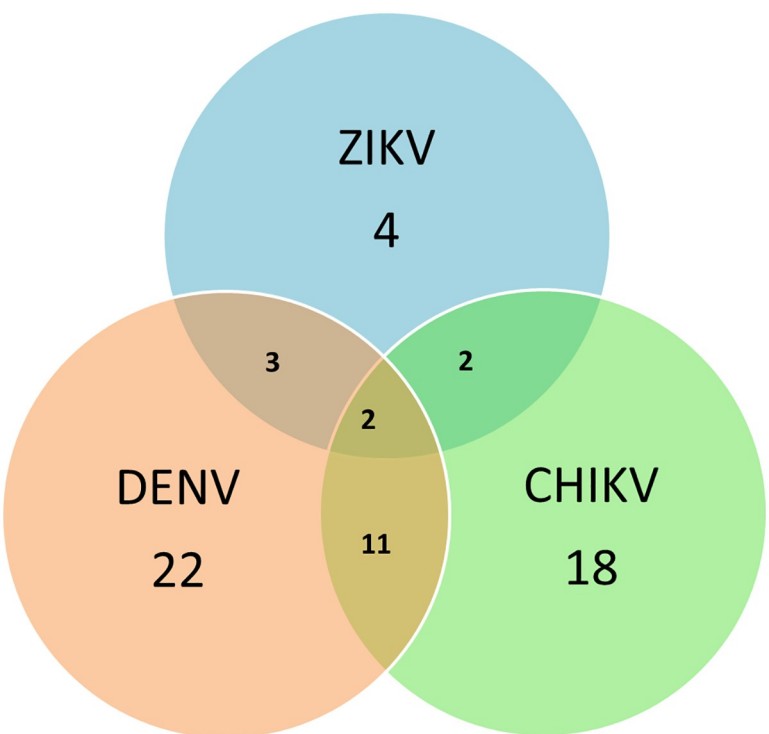

**Fig 4. Graph representing the coinfections that were presented in the analyzed samples.**

and high prevalence of IgG antibodies against the three viruses that was found (83.9% for DENV, 65% for ZIKV and 59.1% for CHIKV), shows that in the analyzed months there was high viral circulation, although cross reactions can be found among antibodies against DENV and ZIKV. This fact is very important because newborns of infected women, especially with ZIKV, can present neurological health problems that can be disabling [37].

Epidemiological information is obtained based on the notification of suspected, probable or confirmed cases, as long as clinical symptoms occur, however this information is partial due to the high frequency of asymptomatic infections, so the importance of establishing proactive systems of more stringent epidemiological surveillance, in which studies are complemented in an apparently healthy population, without recent symptoms and highly susceptible to be able to obtain accurate information from the molecular and immunological point of view and thus calculate the proportion of the population that could have recently been infected and what is a repeat infection.

Asymptomatic infections could explain that people living in a defined endemic region can re-become infected, leading to silent infections with a positive laboratory diagnosis and that

**Table 2. IgG results of recent infections of pregnant woman study and serological incidence of Dengue, Zika and Chikungunya.**

| Age groups | total | IgG anti-DENV | % | IgG anti-ZIKV | % | IgG anti-CHIKV | % |
|---|---|---|---|---|---|---|---|
| <20 years | 33 | 30 | 88.2 | 23 | 67.6 | 22 | 64.7 |
| 20–24 | 41 | 35 | 29.0 | 25 | 61.0 | 9 | 22.0 |
| 25–29 | 34 | 28 | 82.4 | 21 | 61.8 | 18 | 52.9 |
| 30–34 | 16 | 13 | 81.3 | 12 | 75.0 | 10 | 62.5 |
| 35 y + | 12 | 9 | 75.0 | 4 | 33.3 | 6 | 50.0 |
| total | 136 | 115 | 83.9 | 85 | 62.0 | 65 | 47.4 |

play an important role in virus preservation and possibly play a role in transmission of the virus by its vectors; in the case of dengue, asymptomatic infections could represent re-infections by homologous serotypes or indicate the possible risk of developing severe forms with heterologous serotype infections.

It is also important to emphasize that when a person is previously infected by a specific virus, he develops specific antibodies against that virus and when he is later infected by the same virus, he assembles an important immune response, that is, they could be interpreted as a natural revaccination.

## Author Contributions

**Conceptualization:** Leticia Eligio-García, Jorge Fernando Méndez–Galván, Enedina Jiménez-Cardoso.

**Data curation:** Leticia Eligio-García, María del Pilar Crisóstomo-Vázquez, Jorge Fernando Méndez–Galván, Enedina Jiménez-Cardoso.

**Formal analysis:** Leticia Eligio-García, María del Pilar Crisóstomo-Vázquez, Jorge Fernando Méndez–Galván, Enedina Jiménez-Cardoso.

**Funding acquisition:** Jorge Fernando Méndez–Galván, Enedina Jiménez-Cardoso.

**Investigation:** Leticia Eligio-García, María del Pilar Crisóstomo-Vázquez, María de Lourdes Caballero-García, Mariana Soria-Guerrero, Sury Antonio López-Cancino.

**Methodology:** Leticia Eligio-García, María del Pilar Crisóstomo-Vázquez, María de Lourdes Caballero-García, Mariana Soria-Guerrero, Sury Antonio López-Cancino.

**Project administration:** Jorge Fernando Méndez–Galván, Enedina Jiménez-Cardoso.

**Resources:** Sury Antonio López-Cancino.

**Software:** Leticia Eligio-García.

**Visualization:** Leticia Eligio-García.

**Writing – original draft:** Leticia Eligio-García, María del Pilar Crisóstomo-Vázquez, Jorge Fernando Méndez–Galván.

**Writing – review & editing:** Leticia Eligio-García, Jorge Fernando Méndez–Galván, Enedina Jiménez-Cardoso.

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
