## [Decision Letter · Decision Letter 0]

6 May 2020

Dear Ph D Jimenez,

Thank you very much for submitting your manuscript "Co-infection of Dengue, Zika and Chikungunya in a group of pregnant women from Tuxtla Gutiérrez, Chiapas: preliminary data. 2019" for consideration at PLOS Neglected Tropical Diseases. As with all papers reviewed by the journal, your manuscript was reviewed by members of the editorial board and by several independent reviewers. In light of the reviews (below this email), we would like to invite the resubmission of a significantly-revised version that takes into account the reviewers' comments. 

We cannot make any decision about publication until we have seen the revised manuscript and your response to the reviewers' comments. Your revised manuscript is also likely to be sent to reviewers for further evaluation.

Sincerely,

Paulo Pimenta

Deputy Editor

Reviewer's Responses to Questions

**Key Review Criteria Required for Acceptance?**

**Methods**

-Are the objectives of the study clearly articulated with a clear testable hypothesis stated?

-Is the study design appropriate to address the stated objectives?

-Is the population clearly described and appropriate for the hypothesis being tested?

-Is the sample size sufficient to ensure adequate power to address the hypothesis being tested?

-Were correct statistical analysis used to support conclusions?

-Are there concerns about ethical or regulatory requirements being met?

Reviewer #1: The described methods are sound and rationale is clear.

Reviewer #2: The methods describe in the manuscript was not clear enough and not well organized.

In the abstract authors only describe that the tests for DENV, ZIKV, and CHIKV were qRTPCR, but ini the text there was also ELISA test for IgG for all the three viruses that was not mentioned in the abstract and summary. If authors want to keep the ELISA test presented in the paper, they should change the title or at least abstract and summary to reflect it.

In use of citation in the method section was also not well. Authors described their methods using a methods in other publication, instead of directly refer to what they were performed.

Again the PCR amplication section, authors stated that viral nucleic acid will be amplified using qRTPCR. It better to menttion that the kit they are using was 'one step' RTPCR kit, so reader would understand that there were no different assay which separate the RT step and the amplification step, but simultaneously performed in continue reaction in the same tube.

There were no detail explanation on the PCR mix composition by only mention the template volume 9ul with final volum 15ul. It is necessary to wrote the detail composition the PCR mix, since it is a qRTPCR reaction using probes that each reaction should contain template, dNTP, 3 pairs of primers, 3 probes, RT enzym, tag polymerase and water. it was imposible to mix all the reagents within 6ul since 9ul was used for template with reaction volume only 15ul.

The sequence of the primers and probes also need to be written down and the papers they cited for the primers.

Reviewer #3: （1）The pregnant women in this study were included at random, please provide detailed sampling methods.

（2）In this study, 136 pregnant women aged from from 14 to 43 years old who lived in the central region of the state of Chiapas, Mexico were selected. Whether the subjects (pregnant women) who live in the central of the city have different rates of infection from those in the surrounding areas?

（3）Whether the sample size in this study is enough.

**Results**

-Does the analysis presented match the analysis plan?

-Are the results clearly and completely presented?

-Are the figures (Tables, Images) of sufficient quality for clarity?

Reviewer #1: The results are well-presented, though some of the figures could use improvement.

The data presented in Figure 3 are confusing as an X-Y graph with sample number versus RFU. The authors may consider presenting these data as a scatter plot to better show the range and distribution of their data.

Reviewer #2: The results is an important data for public health that show the rate of asymptomatic infection by the three arboviruses.

It show the evidence of asymptomatic infection in pregnant women in inter epidemic period, with no clinical case reported for ZIKA and Chikungunya. 

The ELISA data was also important for awarness of the arbovirus exposure in population, and should also be highlighted in summary and abstract.

Reviewer #3: It has been observed that the prevalence of primary infections by DENV increases with age (line113-115). We didn't find similar phenomenon in this study (In Table 2). There may be differences between different age groups. Please give some statistical analysis or explanation.

**Conclusions**

-Are the conclusions supported by the data presented?

-Are the limitations of analysis clearly described?

-Do the authors discuss how these data can be helpful to advance our understanding of the topic under study?

-Is public health relevance addressed?

Reviewer #1: The authors should consider improving their discussion - see comments below.

Reviewer #2: Authors concluded a high frequency of asymptomatic infection using the results of qRTPCR and ELISA, but should consider that what was presented by ELISA results may express historical infection that beyond the study period since it detect the IgG that may persist 1-2 year in the serum, while evidence of asymptomatic ZIKA (which should be the main focus on pregnant women, which is only 8%. RT PCR should represent the current distribution of pathogen while ELISA represent historical exposure of the pathogen.

The authors also concluded that re-infection in the form asymptomatic and positif serology may preserve the viruses. It should be interpreted that previous infection that lead to seroconversion will actually protect people from re-infection (except for dengue with heterologous infection), thus the high seroconversion in population is more sign of protection than virus preservation. However, I agree that an active surveilance to include healthy people should be incorporate in the health system to monitor the distribution of arboviruses.

Reviewer #3: (No Response)

**Editorial and Data Presentation Modifications?**

Reviewer #1: Minor grammatical errors throughout should be corrected.

Reviewer #2: Line 36: All authors declared no conflict of interest

Line 38: arboviruses

Line 39-40: endemicity should not be characterized by the high presence of vectors by the incidence or prevalence of a diseases in sequential years.

Line 44: The main objective/purpose of the study

Line 46-49: should also highlight the ELISA results that reflect the exposure history of the viruses

Line 50: what elevated epidemic transmission means?

Line 50: How the data show the presence of an outbreak?

Line 62: the sentence was not reflected in the results and discussion section, particularly clinical manifestation and congenital damage, and should be omitted from the sentences if not discussed.

Line 66: 60-70% asymptomatic infection was not reflected in qRTPCR results, only in ELISA results which was not a presentation of asymptomatic infection but historical infection within 1-2 years and presence of IgG was not a risk for transmission but more act as protection to new infection.

Line 97: vectors

Line 112-113: the citation was only refer to one study not several studies.

Line 115-120: consider rewording 

Line 120-123: consider rewording

Line 126: study with 17-20 respondents is not representative for an epidemiological study

Line 127-132: consider rewording

Line 133: CHIKV and ZIKV were reduced but this did not mean disappearing

Line 135-139: consider rewording

Line 140: consider change “the main of this study” to “the objective of this study”

Line 159-160: written consent from children (under 18 years old) should been given by their parents or a close family, not by them self.

Line 162-166: consider rewording, with no number at the beginning of new sentences.

Line 166: since the viral samples are RNA virus, was it OK to store it in -20oC before use?

Line 174-175: please make sure whether the extracted RNA was first used to synthesize cDNA, and later amplified by PCR or directly put in one continues reaction of one step qRTPCR?

Line 177-180: please reconfirm this paragraph since it was a qRTPCR (one step reaction probably) that need template, dNTP, RT enzyme, PCR tag polymerase, 3 pairs of primers, 3 probes and water. How the reaction volume can be only 15 ul if the template itself was 9 ul. It means that only other substrates was packed in 6 ul including water?

Line 184-185: Since it was multiplex qRTPCR where primers and probes of all three viruses were mixed, how authors distinguished among DENV, CHIKV and ZIKV if only use one dye?

Line 185-188: The method lack the list of primers and probes sequences and the citation of the primers and probes.

Line 192-209: The ELISA methods and results were not highlighted in abstract and summary.

Line 235: I do not think this graph is important to show, instead we need to know how the DENV, CHIKV and ZIKV were distinguished from one tube reaction, and how the average viral load for each virus.

Line 237-242: This was an important epidemiological data that did not highlighted in abstract and summary.

Line 261-264: consider rewording

Line 268-269: Probably this should mean the sampling period preceded the transmission season of 2019

Reviewer #3: (No Response)

**Summary and General Comments**

Reviewer #1: In this short article by Eligio-García and colleagues, the authors investigate the prevalence of Zika, dengue, and chikungunya viruses in Chiapas, sampling from 136 pregnant women in the region. The authors find that a large portion of these women are either qPCR or IgG positive for these viruses, either alone or in combination. These findings are interesting and provide insight to the epidemiology of these viruses in Chiapas. The authors nicely introduce the topic and describe their work, and their methodology seems sound. My main concern is that this data reads as very preliminary. Correlating these infections with clinical outcomes would significantly enhance the work, though it is unclear if this information is yet available. 

The introduction is thorough and nicely describes the situation with dengue, Zika, and chikungunyaviruses. The authors should include a paragraph at the end summarizing their findings. Perhaps some details about numbers of patients and virus prevalence found in their study belongs in the introduction and in the abstract.

In the discussion, the authors briefly mention co-infection. Some additional information about the clinical implications of co-infection would add to their discussion significantly. They do mention that this has implications for virus circulation.

Is information available concerning ZIKV infections and microcephaly or other neurological issues? Consideration of this data would be helpful in understanding the impact of ZIKV prevalence in their study.

The authors allude to antibody dependent enhancement on lines 293-295. They should consider describing this phenomenon more fully.

Reviewer #2: The data presented by authors was actually important for public health awareness. However, the way authors present it was not clear enough. Including: 

The use of English that did not follow common rules, such as directly wrote down number (instead of letters/words) in the beginning of new sentences.

The background did not describe the problem of arboviruses in pregnant women, but more about general population. Just the last sentence of the background suddenly jumped to the assay that would be performed on pregnant women.

Methods were not describe clear enough and lack many details, such as PCR mix composition and volume, and the sequences of primers and probes used.

Figure 1 is unclear and poor in resolution and may be omitted since it was not the study results

Reviewer #3: (No Response)

PLOS authors have the option to publish the peer review history of their article (what does this mean?). If published, this will include your full peer review and any attached files.

Reviewer #1: No

Reviewer #2: Yes: ISRA WAHID

Reviewer #3: No
---

## [Decision Letter · Decision Letter 1]

17 Jul 2020

Dear Ph D Jimenez,

Thank you very much for submitting your manuscript "Co-infection of Dengue, Zika and Chikungunya in a group of pregnant women from Tuxtla Gutiérrez, Chiapas: preliminary data. 2019" for consideration at PLOS Neglected Tropical Diseases. As with all papers reviewed by the journal, your manuscript was reviewed by members of the editorial board and by several independent reviewers. The reviewers appreciated the attention to an important topic. Based on the reviews, we are likely to accept this manuscript for publication, providing that you modify the manuscript according to the review recommendations. 

Sincerely,

Paulo Pimenta

Deputy Editor

Reviewer's Responses to Questions

**Key Review Criteria Required for Acceptance?**

**Methods**

-Are the objectives of the study clearly articulated with a clear testable hypothesis stated?

-Is the study design appropriate to address the stated objectives?

-Is the population clearly described and appropriate for the hypothesis being tested?

-Is the sample size sufficient to ensure adequate power to address the hypothesis being tested?

-Were correct statistical analysis used to support conclusions?

-Are there concerns about ethical or regulatory requirements being met?

Reviewer #1: Acceptable.

Reviewer #2: Authors have addressed all the question and improve the method section

Reviewer #3: The methods describe in the manuscript was clear and well organized, but the expression is too tedious, such as the acquisition of serum samples, centrifugation and serum storage. Please express more concisely.

**Results**

-Does the analysis presented match the analysis plan?

-Are the results clearly and completely presented?

-Are the figures (Tables, Images) of sufficient quality for clarity?

Reviewer #1: The data presented are much clearer than the initial submission and satisfy my comments.

Reviewer #2: Authors have addressed all the question and improve the results section, however the edited figure and its embedded word are still not good enough since the authors used bolded letters that too big for accompany the figure. Please check figure standard requirement by PLOS NTD.

Reviewer #3: The analysis presented match the analysis plan and the results clearly and completely presented.

The findings have great public health value and have certain significance in guiding the construction of a more sensitive monitoring system.

**Conclusions**

-Are the conclusions supported by the data presented?

-Are the limitations of analysis clearly described?

-Do the authors discuss how these data can be helpful to advance our understanding of the topic under study?

-Is public health relevance addressed?

Reviewer #1: The discussion is improved, but the examination of the topic remains at the surface. The inclusion of a paragraph concerning asymptomatic infected individuals is a nice addition, but the discussion ends rather abruptly.

Reviewer #2: Have been improved.

Reviewer #3: (No Response)

**Editorial and Data Presentation Modifications?**

Reviewer #1: (No Response)

Reviewer #2: Please make fine adjustment in the figure so there a balance of figure and the letter used in associated figure, not too thick or too large. Please check figure standard requirement by PLOS NTD.

Reviewer #3: (No Response)

**Summary and General Comments**

Reviewer #1: Overall, the study as presented is in decent shape. Minor grammatical errors remain but are significantly improved from the prior submission. The other reviewers' comments are relatively well-addressed from my perspective.

Reviewer #2: The authors in general have improved the manuscript to make it clearer, only figure clarity and letter balance need to be considered (font sizes and thickness)

Reviewer #3: This study has important public health implications. However, the methodological description should be more concise.

PLOS authors have the option to publish the peer review history of their article (what does this mean?). If published, this will include your full peer review and any attached files.

Reviewer #1: No

Reviewer #2: Yes: Isra Wahid

Reviewer #3: No
---

## [Editor Report · Decision Letter 2]

21 Sep 2020

Dear Ph D Jimenez,

Thank you very much for submitting your manuscript "Co-infection of Dengue, Zika and Chikungunya in a group of pregnant women from Tuxtla Gutiérrez, Chiapas: preliminary data. 2019" for consideration at PLOS Neglected Tropical Diseases. As with all papers reviewed by the journal, your manuscript was reviewed by members of the editorial board and by several independent reviewers. The reviewers appreciated the attention to an important topic. Based on the reviews, we are likely to accept this manuscript for publication, providing that you modify the manuscript according to the review recommendations. 

Sincerely,

Paulo F. P. Pimenta, Ph.D

Deputy Editor

Paulo Pimenta

Deputy Editor
---

## [Editor Report · Decision Letter 3]

13 Oct 2020

Dear Dr. Jimenez,

We are pleased to inform you that your manuscript 'Co-infection of Dengue, Zika and Chikungunya in a group of pregnant women from Tuxtla Gutiérrez, Chiapas: preliminary data. 2019' has been provisionally accepted for publication in PLOS Neglected Tropical Diseases.

Best regards,

Paulo F. P. Pimenta, Ph.D

Deputy Editor

Paulo Pimenta

Deputy Editor

---

## [Editor Report · Acceptance letter]

30 Nov 2020

Dear Dr. Jimenez,

We are delighted to inform you that your manuscript, "Co-infection of Dengue, Zika and Chikungunya in a group of pregnant women from Tuxtla Gutiérrez, Chiapas: preliminary data. 2019," has been formally accepted for publication in PLOS Neglected Tropical Diseases.

Best regards,

Shaden Kamhawi

co-Editor-in-Chief

Paul Brindley

co-Editor-in-Chief
